# Steel Bridge-Coating Systems and Their Environmental Impacts: Current Practices and Future Trends

**Jonathan Ralph Adsetts [1], Nafiseh Ebrahimi [1,*], Jieying Zhang [1], Farzad Jalaei [1]** and **Jamie J. Noël [2]**

1     National Research Council Canada, Construction Research Centre, 1200 Montreal Road, Ottawa, ON K1A0R6, Canada

2     Chemistry Department, The University of Western Ontario, 1151 Richmond Street, London, ON N6A 3K7, Canada

\*     Correspondence: nafiseh.ebrahimi@nrc-cnrc.gc.ca

**Abstract:** Coatings are essential for protecting structural steel bridges from corrosion in harsh environmental conditions. The selection of a suitable coating system can significantly impact the performance and longevity of a bridge as well as its environmental footprint. This study investigates the U.S. and Canada's environmental ISO corrosivity map and the general environments to which bridges are exposed. Additionally, environmental data and road maintenance practices of transportation departments and the use of de-icing salts were investigated to examine the impact of micro-corrosive environments on bridge elements. The study reviews commonly examined coating systems and their expected service life in moderate and highly corrosive environments. This sheds light on factors influencing coating system selection, such as life-cycle cost analysis and maintenance practices for bridge elements. For the first time to our knowledge, an environmental life-cycle evaluation of one of the most commonly used coating systems with theoretical maintenance scheduling for a bridge project's expected service life is presented to encourage the use of a quantitative tool for environmental impact assessment of coatings in terms of global warming potential (GWP). Additionally, perspectives on patented state-of-the-art and future steel-protective technologies and their potential role in bridge engineering are reviewed.

**Keywords:** coating systems; life-cycle assessment; self-stratifying; corrosion; micro environment

## 1. Introduction

Physical barriers generally protect steel in infrastructure regardless of whether that physical barrier is a patina on weathering steel, an invisible oxide consisting of Cr(III) and Cr(VI) on stainless steel, or a protective coating system over carbon steel. This physical barrier generally reduces the diffusion of corrosive elements such as chlorides from the road or sea salt, $H_2SO_4$ and $HNO_3$ generated from $SO_2$ and $NO_2$ pollution, water, and air, preventing them from reaching the metal surface and corroding it further. The type of barrier required depends on the environment since some steel is exposed to harsh industrial and marine conditions, and some is not. In general, in order to prevent corrosion, engineers and designers use either a more corrosion-resistant alloy such as weathering steel or stainless steel or apply a coating to the steel bridge.

Properly using coating systems will prevent or slow the corrosion of exposed steel. Without these coatings, steel can immediately start rusting even before bridge construction is complete. The International Organization for Standardization (ISO) defines the longest coating system lifetime category as more than 25 years, with several lower lifetime categories as well [1]. As bridges are generally designed for 75+-year lifetimes, the bridge coatings will need several maintenance or replacement periods to ensure severe bridge corrosion does not occur. As the average age of bridges in the USA continues to increase (currently at 43 years), decisions on bridge coating systems and maintenance to preserve

the life of steel structures are becoming increasingly important. Furthermore, as the climate changes over the expected service lifetime (ESL) of the bridge (75+ years) or even over the ESL of the coating system (25+ years), previous management decisions must be constantly reevaluated [2,3].

The selection and management of coatings should ideally consider environmental factors that can adversely affect coating systems. These factors include average annual humidity, temperatures, exposure to chemicals such as spills, airborne pollutants from industrial areas, and salt in coastal locations. Guidelines for coating systems, such as ISO 12944, have been developed with these environmental considerations in mind to help ensure that coating selection and management decisions are appropriate for specific environmental conditions [4,5]. In the United States, The American Association of State Highway Officials (AASHTO) has published several books on bridge maintenance and management in collaboration with the National Transportation Product Evaluation Program (NTPEP), which evaluates and recommends products [6]. The Society for Protective Coatings (SSPC), which merged with National Association of Corrosion Engineers (NACE) to become the Association for Materials Protection and Performance (AMPP), also provides specific guidelines for painting and cleaning structural steel [7]. To assist government agencies in selecting appropriate coating systems, the Federal Highway Administration (FHWA) offers technical notes, conducts state-of-the-art research, and provides research facilities. In Canada, the Canadian Standards Association (CSA) Group also provides corrosion protection recommendations based on the environment, similar to ISO 12944, where chloride exposure, wetness, marine exposure, and heavy industrial atmosphere are all considered to form 10 groups in the Canadian Highway Bridge Design Code [4]. They make recommendations such as applying coating, using uncoated weathering steel, and galvanizing, metallizing, and investigating site conditions.

This work focuses on the corrosion protecting of exposed steel bridge components, specifically superstructure components such as girders, trusses, floor beams, and bearings. It is important to note that the information provided may not apply to other steel elements such as signs, handrails, and posts. When it comes to coating systems for steel superstructures, there are a limited number of options currently used in practice, with two- or three-coat systems being the most commonly used. While not all coating systems are suitable for every bridge, there are some general and common performance criteria that every coating ideally satisfies, such as the following:

- Effectively reduces or prevents steel corrosion;
- Simple coating application, maintenance and removal procedures;
- Aesthetically pleasing;
- Low life-cycle costs;
- Low environmental impacts;
- Available to market

Among these performance criteria, some life-cycle cost (LCC) studies have considered the performance of anti-corrosion coatings to arrive at the lower-cost solutions [8]. However, to the best of our knowledge, the carbon emission or global warming potential (GWP) from the coating systems through life-cycle assessments (LCA) or other means has not yet been examined in the literature though a bridge's service life.

LCA can provide credible and standardized environmental profiles of infrastructure projects, assisting decision-making infrastructure engineers in selecting the most ecologically optimal designs. In infrastructure-type LCA studies, including bridge LCA, the overall environmental impacts from the entire life cycle of a bridge are considered [9–12]. However, due to the complexities of environmental concerns and the diversity of bridge construction, conducting a thorough environmental assessment of bridges and bridge-coating systems is far from simple. This study contains an LCA conducted for the most robust and common paint coating system used in North America, with appropriate coating system maintenance scheduling for a bridge ESL of 64 years. The GWP of the transportation and application of coating systems was excluded since these factors are relatively similar between coating

systems; this LCA focuses on the total GWP of the ingredients used in the coating systems over the bridge's ESL.

The authors also review and evaluate the emerging technologies that are expected to impact the field of bridge coatings. These technologies include metalizing faying surfaces, self-stratifying coatings, and two-coating systems incorporating conductive lamellar fillers in the zinc primer layer.

## 2. Environment Corrosivity Classification

Environmental corrosivity identification is essential for implementing corrosion protection and control throughout the life cycle of a bridge. Atmospheric corrosion is one of the primary causes of steel bridge corrosion, as it is affected by various factors such as temperature, humidity, salinity, and pollution levels. These factors can vary within a given location and can change year to year.

ISO has developed a guide called "Corrosion protection of steel structures by protective coating systems", commonly known as ISO 12944. This guide categorizes environments into six distinct categories: C1 through C5 and CX [5], as follows:

- C1—very low: a mild environment with low humidity and no pollution. Corrosion rate of carbon steel is <1.3 μm/year in a C1 environment [5];
- C2—low: a low-corrosion-risk environment with occasional condensation and low pollution. Corrosion rate of carbon steel is 1.3–25 μm/year in a C2 environment [5];
- C3—medium: a moderately corrosive environment with high humidity, occasional condensation, and moderate pollution levels or some effect of chloride. Corrosion rate of carbon steel is 25–50 μm/year in a C3 environment [5];
- C4—high: a high-corrosion-risk environment with constant high humidity, regular condensation, and high pollution levels or substantial chloride effect. Corrosion rate of carbon steel is 50–80 μm/year in a C4 environment [5];
- C5—very high: a subtropical zone with very high pollution and/or exposure to saltwater or other aggressive substances such as chemical plants, offshore structures, or coastal areas. Corrosion rate of carbon steel is 80–200 μm/year in a C5 environment [5];
- CX—extreme: a subtropical and tropical zone with very high pollution and/or strong effect of chlorides with high levels of aggressive substances and environments with extreme temperature or pressure conditions. Corrosion rate of carbon steel is 200–700 μm/year in a CX environment [5].

The ISO corrosivity category can be determined by direct measurement of the one-year corrosion thickness/mass loss of standard specimens exposed to the environment or by the use of corrosion models that require the input of detailed environmental information of a location [13].

In recent years, a hybrid tool using existing modelling and mapping solutions called ISO Corrosivity Category Estimation Tool (ICCET) was developed by the U.S. Department of Defense (DOD). This web-based automated tool combines ISO corrosion estimation models with National Oceanic and Atmospheric Administration (NOAA) environmental data and produces the general ISO corrosivity of the environment [14].

Figure 1 shows the corrosion map and corrosivity category created by ICCET in Canada and the U.S. Additional data were added to the figure for Canadian cities from our previously published article [15]. While most inland locations show a C1–C2 corrosion category, locations near salt water show corrosion categories of C4 to C5, emphasizing the negative effect of humidity and chloride on corrosion [16]. The salt deposition is also common within 10 kilometers of a sea due to the formation and deposition of salt-bearing aerosols. It has been reported previously that with increasing distance from waterlines, the ISO category can drop from CX to C4 [17]. Compounding the inherent salt risk is salt's hygroscopic nature, which can result in the formation of liquid droplets from humid air.

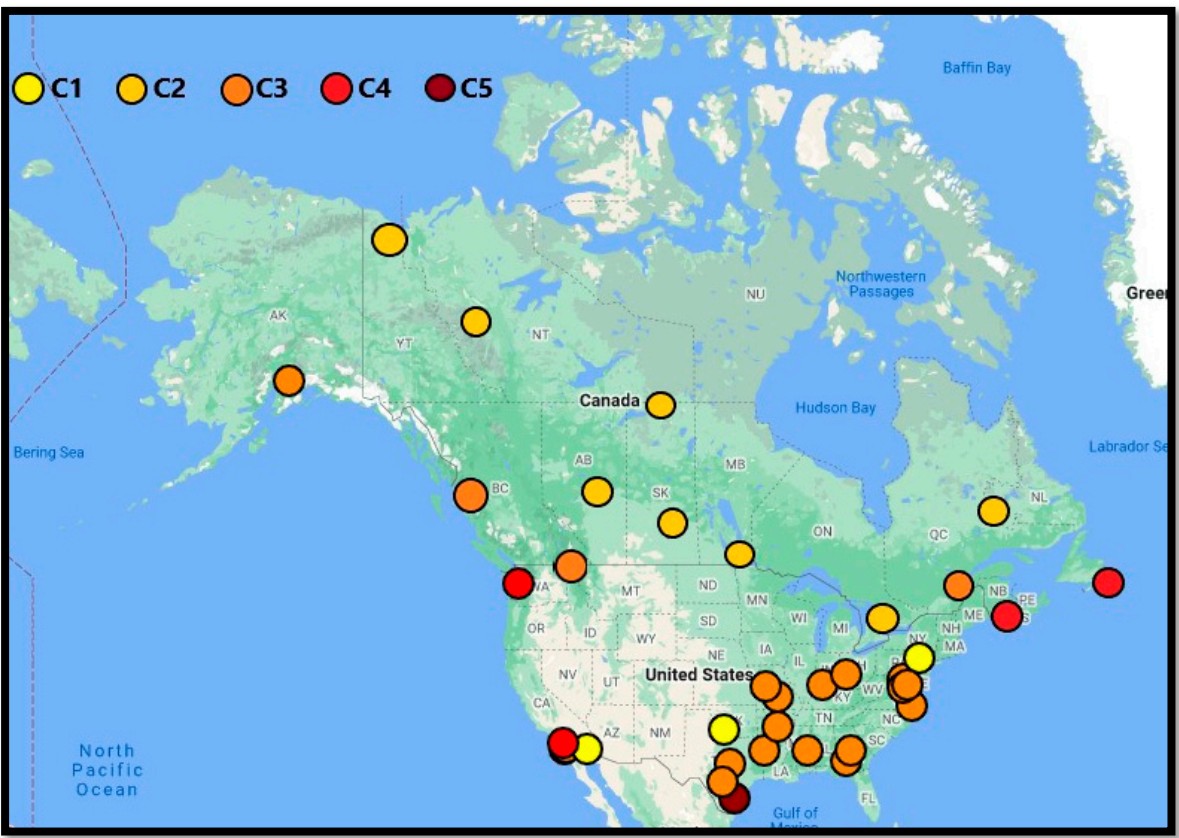

**Figure 1.** ISO Atmospheric corrosivity map of USA and Canada based generated by ICCET; additional data were added to the figure for Canadian cities from our previously published article [15].

While environmental severity classification is a valuable tool for overall environment corrosivity estimation and definition of the macro environment, it is not indicative of absolute corrosion potential or the total environmental corrosivity of a bridge on the micro level. The actual environment that affects a specific material or system correlates directly to the conditions of the micro-environment that it experiences. There may be significant variations in environmental parameters that influence corrosion, i.e., humidity, chloride, and pollution, on some parts of a bridge that differ from the conditions at other locations on the same bridge. Examples of highly corrosive micro-environments on a bridge include highway crossings with de-icing salt use, water crossing with low vertical clearance, and sites with dense vegetation or shelter.

To assess the micro-corrosivity of infrastructure, we focused on two key factors: relative humidity and chloride presence, as the environmental laws have significantly reduced the pollution levels in environments [15]. The average hourly relative humidity over a year for most of North America was collected from the National Centers for Environmental Information as well as road management practices from all transportation agencies in the USA. The use of de-icing road salt in Canada was also considered. This information allowed us to categorize locations based on the risk of establishing a corrosive micro-environment. Locations with an average relative humidity of 70% were defined as high-humidity since road de-icing salts have a minimum deliquescence point at 76% relative humidity [18], which means a salty, corrosive solution will form spontaneously in these environments. Figure 2 illustrates the color-coded regions based on our analysis, which yielded the following:

1. Regions with average annual humidity under 70% and no road salt used (green);
2. Coastal regions with no road salt usage (yellow);
3. Coastal regions with road salt usage (orange);
4. Regions with average annual humidity greater than 70% with road salt used (red).

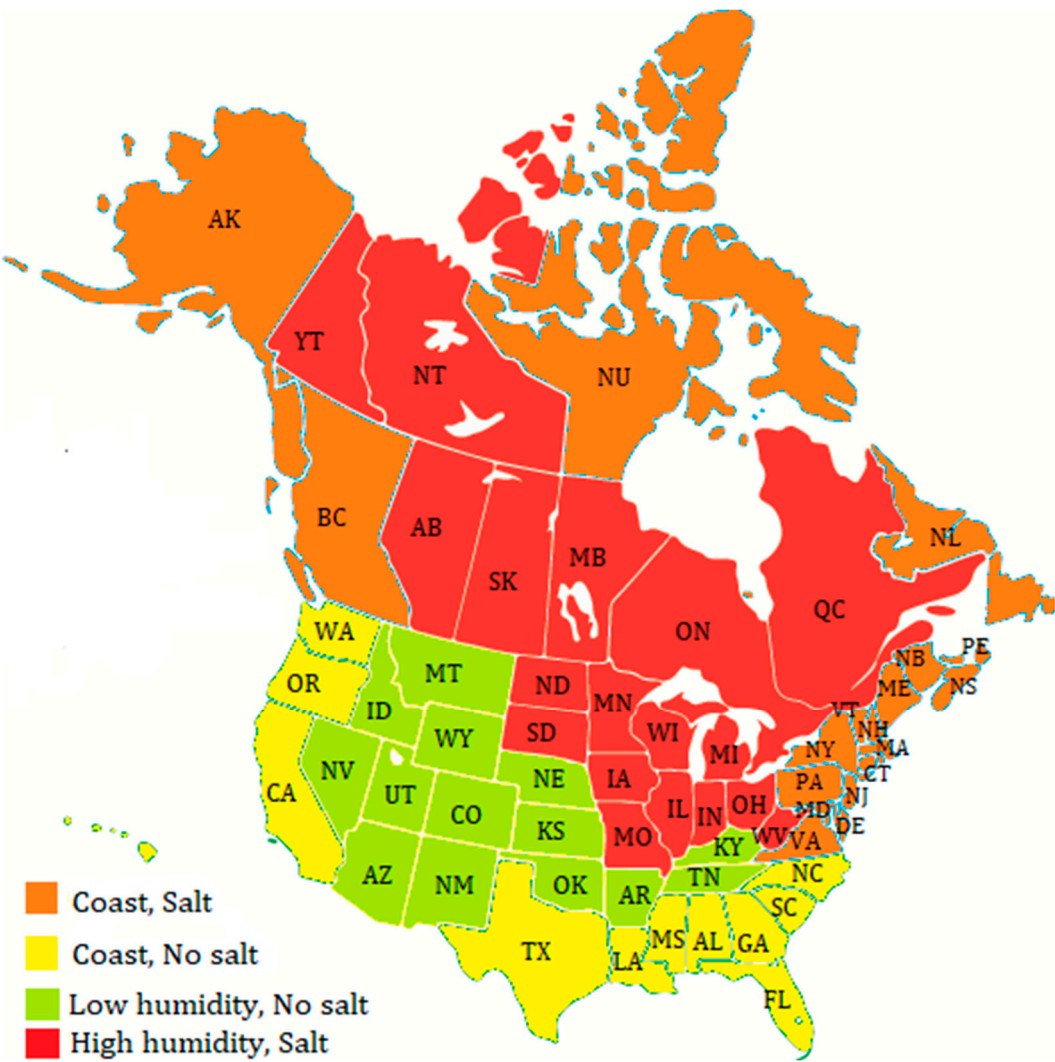

**Figure 2.** Grouping of states/provinces/territories by most extreme environmental condition for corrosion.

Red regions pose the greatest risk of corrosion. In Canada, these regions include the Maritimes, BC's coast, Nunavut, Newfoundland and Labrador, Ontario, and Quebec. The USA has also experienced severe corrosion of steel materials in similar environments. In fact, in 1989, Detroit led Michigan to establish a moratorium on the use of all types of weathering steel in construction because of its rapid corrosion rate [19]. Furthermore, according to a 2014 paper in the National Review on Use and Performance of Uncoated Weathering Steel Highway Bridges, all states touching the Great Lakes experienced corrosion problems with uncoated weathering steel bridges [20]. Comparatively, uncoated weathering steel is predicted to last up to 75 years in some drier continental states in the green or yellow zones [21].

Infrastructure protection against corrosive macro- and micro-environments is vital to safeguarding against the cumulative impact of these environmental factors. Choosing the appropriate coating system based on both macro- and micro-environments is also crucial to ensuring optimal protection and preventing severe corrosion damage.

## 3. Coating Systems Selection

There are various options available for corrosion protection of structural steels. When the environment is favorable, the use of uncoated weathering steel is recommended. Its corrosion resistance comes from alloying steel with small amounts of copper, nickel, chromium,

silicon, and phosphorus. In such an environment, the oxides that form on the weathering steel, i.e., the patina, adhere tightly to the surface and protect it from corrosion. However, the patina is unstable in corrosive environments and may need extra thickness or coating. Some departments of transportation (DOTs) recommend zone painting of the most corrosive area of uncoated steel to limit painting and maintenance of their assets, for example, painting the end of steel members below joints [22].

A 2018 survey by the Transportation and Research Board in the USA found that new U.S. bridges use zinc-based coatings 43% of the time, metallized and galvanized steel 6% each, non-zinc coatings 3%, and uncoated weathering steel 42% of the time [22]. According to the NEPCOAT website, a large number of state highway departments mandate the use of zinc-rich primer coatings. Zinc-rich coatings usually consist of two or three layers of paint, with the primer coat containing a significant amount of zinc pigment for cathodic protection. These primers come in two varieties: inorganic zinc (IOZ) and organic zinc (OZ). IOZ primers are comprised of zinc metal powder blended with an inorganic silicate paint binder, which can be either solvent-borne (ethyl silicate) or water-borne (alkali silicate). OZ primers, on the other hand, contain zinc metal pigment mixed with an organic paint resin such as epoxy or urethane. They are designed to provide good adhesion for the application of subsequent layers.

A brief explanation of how the three-coat zinc-rich coating system fails in the environment is displayed in Figure 3. The layers degrade from the outermost layer down to the bottom. The coating system has a topcoat made of polyurethane, a midcoat consisting of a generic polymer, and a bottom layer consisting of over 80% weight zinc in a dry film mixed with epoxy.

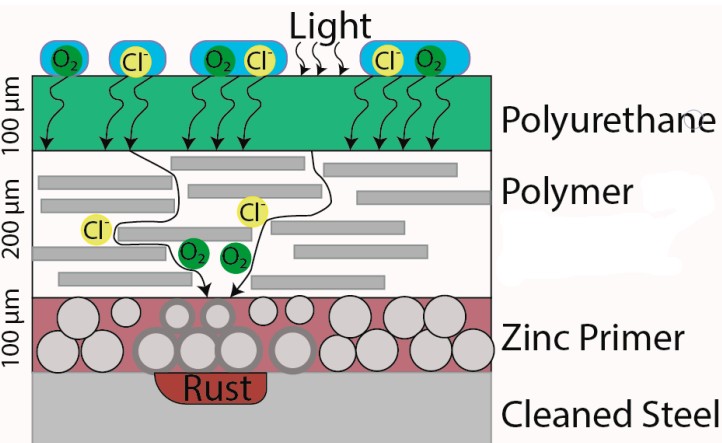

**Figure 3.** Basic coating failure mechanism for the three-coat zinc-rich coating system. This system consists of three layers with a sacrificial zinc primer on the steel's surface, a barrier polymer as a midcoat, and a polyurethane topcoat.

The top layer functions as a hydrophobic layer that repels water by creating a low contact angle with water. This lower water–surface contact slows the diffusion of ions or gases into the topcoat. The topcoat also absorbs light (hv) to prevent the bottom layers from deteriorating. This layer is malleable, resists abrasions, and can spread over the surface of the midcoat. However, its malleability allows ions and gases to diffuse through.

The midcoat is generally a highly crystalline, rigid polymer that is below the polymer's glassy temperature (GT). These polymers generally consist of polyesters and polyamides. The midcoat prevents the flow of ions and gases diffusing through the midcoat to slow down the rate of corrosion. Due to this polymer's crystalline nature, it often fails through delamination or deadhesion.

The bottom layer contains mostly zinc and some filler. This filler can be organic as an epoxy or inorganic as a silicate. The zinc metal is higher on the galvanic series than iron and therefore preferentially corrodes instead of the iron. When Zn is corrodes, it expands

dramatically. This expansion may break the highly crystalline midcoat and lead to diffusion of more corrosive elements to the substrate. Once zinc is consumed, the steel beneath begins corroding, and FeO flakes can separate from the steel and fall off with the coating, leading to coating failure.

Table 1 lists typical corrosion protection coatings used in infrastructure and their practical maintenance times according to the recently published work by Helsel et al. [23]. Practical maintenance time is the time until 5 to 10% coating breakdown occurs, and an active rusting of the substrate is observed.

**Table 1.** Commonly used coating systems in infrastructure and their maintenance times in various environments. All three-coat systems consist of the following: coat 1: organic or inorganic zinc-rich primer; coat 2: epoxy or urethane intermediate coating; coat 3: polyurethane or acrylic latex topcoat. Two-coat systems consist of a zinc-rich primer followed by a polyaspartic, polysiloxane, or acrylic finish.

| Corrosion Protection System | Specification | Coating System (Primer/Midcoat/Topcoat) | Practical Maintenance Time in Years in the Environment [23] |
|---|---|---|---|
| Zinc-based coatings | Organic zinc primer | 2-coat system (OZ/polyaspartic) | 18 (C3), 14 (C5) |
| | | 3-coat system (OZ/epoxy/polyurethane) | 21 (C3), 13 (C5) |
| | Inorganic zinc primer | 2-coats system, (IOZ/epoxy) | 18 (C3), 14 (C5) |
| | | 3-coat system (IOZ/epoxy/polyurethane) | 21 (C3), 16 (C5) |
| Nonzinc coating system | Alkyd systems | 3-coat system alkyd/alkyd/urethane alkyd | 13 (C3), 9 (C5) |
| Metallizing | 85% zinc, 15% aluminum | Metallizing | 22 (C3), 16 (C5) |
| | | Metallizing/sealer | 25 (C3), 18 (C5) |
| | | Metallizing/sealer/polyurethane | 28 (C3), 22 (C5) |
| Hot-dipped galvanizing | 0.004 inch minimum galvanizing | 1-coat system | 90 (C3), 72 (C5) |

For zinc-based coatings, the practical maintenance time in C3 environments ranges from 18 to 21 years depending on the number of coats and 13 to 16 years in C5 environments. The IOZ/epoxy/polyurethane coating system has a practical maintenance time in a C5 environment compared to other zinc-based coatings. Generally, IOZ primers are used in high-salt-content environments, making their unanimous usage in coastal regions obvious. However, they are much harder to apply and cure properly; therefore, most provinces/regions and territories only apply IOZ primers in shop settings and use OZ primers in the field.

Non-zinc coating systems are less commonly used and are mostly restricted to non-aggressive environments; as seen in Table 1, the alkyd system requires maintenance in 13 years in C3 and 9 years in C5 environments, which provides much lower protection than zinc-based coatings. Other available options are metallizing and galvanizing. Metallizing systems typically use a thermal spray to apply a layer of 85% zinc and 15% aluminum on the metal surface. The metallized surface can be further coated with a sealer and polyurethane topcoat for enhanced corrosion protection to increase the practical maintenance time of coating to 28 years in C3 and 22 years in C5 environments. Hot-dipped galvanizing involves immersing the metal structure in molten zinc to form a coating on the surface. Hot-dipped galvanizing is found to provide superior corrosion protection and has the highest practical maintenance time in C3 and C5 environments, ranging from 72 to 90 years in C3 and C5 environments, respectively.

Typical maintenance practices of the coating after the original painting include (1) spot touch-up for one or two cycles at the practical maintenance time, (2) maintenance repair (spot prime and full coat), and (3) total coating removal and replacement [17,24]. Preventative maintenance can greatly extend the lifetime of a coating if performed at the right time. When less than 10% of the surface is corroded, spot touch-up maintenance is adequate, while if heavy coating breakdown occurs, and rust is visible on over 20% of the surface, full recoating is required. The time available to perform preventative maintenance or touch-ups can be short, as the corrosion spreads quickly once initiated. The time between 5%–10% surface rusting and severe corrosion product build up and coating delaminating is roughly 6–7 years [3]. Waiting until coating system failure is generally not economical, as the full recoating cost can be over two times the original coating cost [23].

When selecting a coating system, life-cycle cost analysis is an excellent decision-support tool for bridge owners to choose the most cost-effective coating system for their specific environment. Surface preparation, cost of operation, waste containment, and disposal and technical costs of frequent maintenance should be considered. Table 2 shows an example of a performance matrix for a bridge-coating system selection and their coating systems' cost in USD per square meter. It includes labor, equipment, consumables, materials, and waste containment, based on class 2 systems. As seen in Table 2, for a highly complex structure, it is more feasible to choose a metallizing option with higher expected service life despite the higher initial cost of coating since the cost of maintenance and waste containment is high due to altitude and the complexity of the structure; therefore, for the designed service life of the structure, the total life-cycle cost is lower. On the other hand, for a low-altitude, simple structure with hand-cleaning waste containment of old paint, an alkyd coating system may be more cost effective.

**Table 2.** Performance matrix for selecting bridge-coating systems. All costs are in USD per square meter and include labor, equipment, and consumables, where job cost modifiers are included in parentheses. All values are for CX environments and for field applications. All coating containment strategies are based on class 2 containment systems[2,4].

| Ranking | 1 (Low) | 2 | 3 | 4 (High) |
|---|---|---|---|---|
| Bridge Accessibility | Simple structures < 15 m high (×1.20) | Complex structures < 15 m low (×1.35) | Simple structures > 15 m high (×1.45) | Complex structures > 15 m high (×1.50) |
| Approx. Desired Service Lifetime | 5 years (hand- or power-clean/alkyd primer/alkyd midcoat/alkyd topcoat, USD 12.48 + USD 5.44 + USD 2.50 + USD 1.72 = USD 22.14) | 9 years (blast-clean/epoxy phenolic primer/epoxy ester topcoat, USD 22.92 + USD 5.28 + USD 1.98 = USD 30.18) | 15 years (blast-clean/IOZ/epoxy/polyurethane sealer, USD 22.92 + USD 5.37 + USD 3.34 + USD 2.56 = USD 34.19) | 22 years (SP-10 cleaning/metallizing/sealer/polyurethane, USD 22.92 + USD 109.42 + USD 6.78 + USD 2.56 = USD 141.68) |
| Old Waste Coating Containment Strategies | Hand-cleaning simple structure < 15 m (×1.75) | Hand-cleaning complex structure < 15 m (× 2.50) | Blasting simple structure > 15 m (×2.75) | Blasting complex structure > 15 m (×3.50) |

Not included as considerations for the coating selection are the environmental carbon costs for using certain chemicals in coating and the social costs required for bridge shutdowns for maintenance. The following section reviews the environmental impacts of certain coating systems over a bridge's ESL.

## 4. Life-Cycle Assessment for Coating Systems' Environmental Considerations

Not at all coating system management decisions must be purely financial; the environmental impacts of construction projects have recently garnered more scrutiny [10]. This section provides a brief insight into predicting a coating system's environmental impact over the lifetime of a specific bridge. To the best of our knowledge, this is the first LCA study performed on a coating system for a bridge.

## 5. Case Study Details and Assumptions

For this LCA study, an example bridge is used, in which all bridge element information came from the USA's NBI. The USA's Corp of Engineers (civil) own a bridge at the Kentucky/Ohio border that is 1075 m long and 13.1 m wide, labelled CEPORHKY4028601 in the NBI. However, this bridge lacks the element data necessary to complete an LCA, so element information from a Kentucky bridge of similar length and width (079B00151N) was used. This bridge is slightly shorter at 950 m but wider at 15.2 m wide, with 35,303 $m^2$ of exposed steel girder/beam superstructure, which requires a coating system. The surface area of the exposed steel girder/beam superstructure is assumed to be the same for both bridges.

For the specific coating system, Kentucky's approved product list has type I, class V products for their bridges, which consist of an IOZ primer, an epoxy midcoat, and a polyurethane topcoat [25]. The specific three coats chosen for analysis were Carbozinc 11 HS (IOZ primer), Macropoxy 646 (epoxy midcoat), and Acrolon 218 HS (polyurethane topcoat). This type of coating system is robust and has the longest ESL in any environment [24]. All ingredient information was taken from the safety data sheets (SDS) provided by the manufacturers. Generally, SDS provided ranges for ingredients where the center of the compositional range was always assumed. For example, an ingredient with a compositional range of 10%–30% would be assigned a 20% compositional percentage. Notably, the percentages did not sum to 100%, and the SDSs did not list every ingredient used in the paint.

Kentucky also uses road salt and has average annual relative humidity over 70%, likely placing the environmental zone near a C5/CX environment. All lifetime assumptions are made based on a C5/CX environment with ESL and coating management decisions assumed from data presented by Helsel et al. [24]. It is assumed that bridge is designed for service life of 64 years, and the maintenance schedule was selected so that a touchup will occur at 17 years, then a maintenance at 23 years, and a full repaint at 32 years after initial painting. This maintenance cycle occurs twice and excludes the final repaint.

LCA analysis requires an extreme level of data intensity assumption at each stage of the bridge's service life. These assumptions would be transferred to the bridge data needs as the required input data to conduct the life-cycle environmental impacts analysis. The ecoinvent v3.8 database [26] was used as the background system, while all unit activities representing the foreground system were directly integrated with ecoinvent v3.8, and openLCA 1.10.3 [27] was selected as the engine for LCA calculations. ReCiPe2016 (hierarchical) midpoint impact categories were considered for the production and maintenance of this coating system, including the climate change potential (known as global warming potential—GWP), metal-depletion potential (MDP), fossil-depletion potential (FDP), water-depletion potential (WDP), and particulate matter formation potential (PMFP) [28].

Selection of an appropriate functional unit (FU) for an attributional LCA study of infrastructure systems such as bridges is still a challenge. The commonly used unit of 1 $m^2$ (with consideration of the bridge area in which to apply coating system) was selected as the FU for the LCA results. This LCA analysis is only limited in the scope of production of coating ingredients. It includes the following: the upstream processing of materials used for the construction and maintenance of the bridge, starting from raw material extraction to the final product ready to be used; the maintenance schedule; and the various life-cycle activities and processes throughout the life of the bridge structure considered in the planning and design stage of coating systems during the bridge's 64-year life cycle. It does not consider the installment or removal processes of the coatings. Therefore, the GWP (kg $CO_2$ eq.) is illustrated in the LCA results per functional unit ($m^2$) of the coating system.

## 6. Case Study

Figure 4 shows the LCA analysis results per mixed paint part for each layer of the coating system.

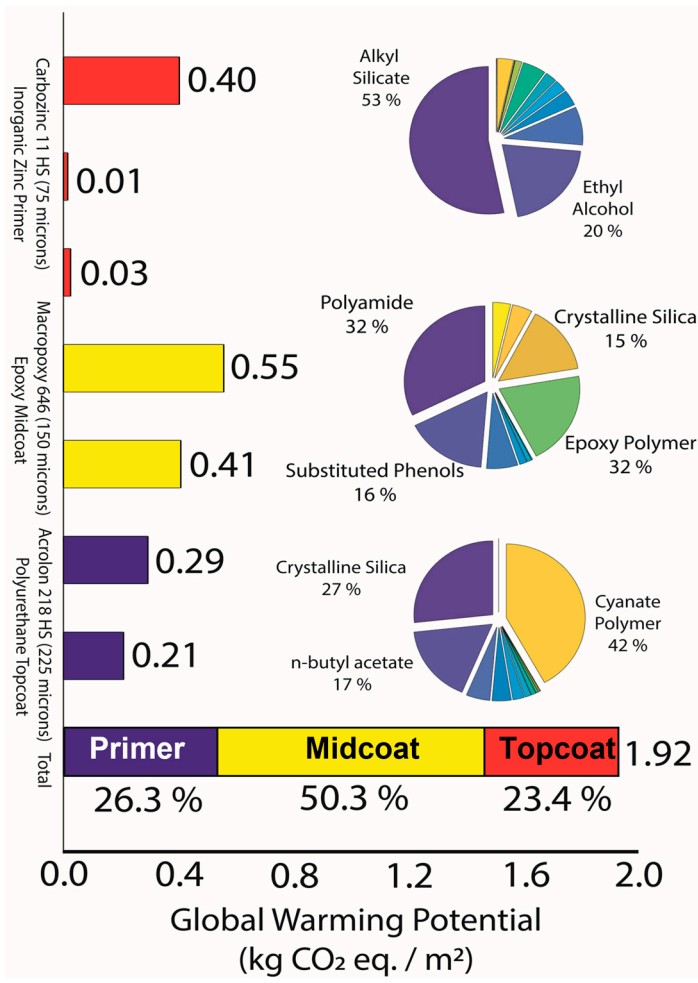

**Figure 4.** Life-cycle impacts assessment (LCI) results (climate change potential in kg $CO_2$ eq./m$^2$ for a robust three-coat system). Each ingredient in each part to be mixed for each layer of the coating system was individually analyzed, where data from the SDS provided by the coating company were used for ingredients. The main individual ingredients and their percentages in each layer are represented in the pie charts, and details of all ingredients can be found in the company's SDS.

Clearly, the epoxy midcoat contains the highest GWP of the film at over 50% of the total GWP. Over one-quarter of the GWP from the entire coating system was from the polymers in the midcoat, namely polyamide and epoxy polymers. These polymers consist of 18.75% of the weight of the layer but have a much higher contribution to the GWP than other components. Unfortunately, these polymers are essential to the function of the coating system. Lowering the $CO_2$ of the midcoat requires higher procurement of low-carbon polymers. One environmental consideration could be to remove this midcoat altogether while sacrificing 3–4 years of ESL for the coating system [24]. Lowering the ESL would mean the layer must be replaced more frequently, while human error is reportedly the largest source of coating system failure [29].

Interestingly, when the GWP is normalized by the layer thickness, the IOZ primer layer GWP becomes comparable to that of epoxy midcoat. The main $CO_2$ contributor in this IOZ primer is the alkyl silicates that give the IOZ primer its "inorganic" nature. In fact, the alkyl silicates have more GWP than the rest of the components in this IOZ primer layer combined. Reducing the GWP of the zinc primer would start with investigating these alkyl

silicates. Surprisingly, the zinc dust and zinc oxide that make up 84% of the weight of the dried IOZ primer contribute less than 1% of the entire coating system GWP and less than 4% to the IOZ primer layer GWP.

As for the polyurethane topcoats' GWP impact results, they are closely tied to the zinc primer layer for the total system $CO_2$ contribution. However, accounting for the thickness of layers, the topcoat contributes three times less than the zinc primer and the midcoat. Similar to the other layers discussed so far, crystalline silica and cyanate polymers contribute to 70% of this layer's GWP. Overall, silica-containing compounds and polymers contribute to 65% of the entire system GWP. Notably, these compounds comprise, on average, roughly 38% of the weight of the entire system before the paint is applied to the bridge or any drying of the coating system occurs.

When considering the whole life cycle of bridges and the amount of exposed steel that must be protected, large GWP contributions from the coating systems are to be expected, whereas few if any studies have considered this environmental impact. Considering that some bridges can have thousands of tons of coating systems applied over their ESL (e.g., the Golden Gate Bridge), use of a low-carbon coating system can significantly reduce the environmental impact. It should be noted that the coating system's GWP impacts are normally estimated to be around $\leq$1%–2% relative to the whole-bridge LCA results depending on a variety of factors that could directly and indirectly affect this comparative analysis (i.e., inclusion of whole-bridge life-cycle stages in the LCA study, materials waste, transport fleet-related impacts in maintenance and rehabilitation of the bridge, inclusion of chloride-induced corrosion of bridge decks, etc.). Transportation agencies and decision makers would benefit from a database containing environmental impact assessments of coating systems to help make environmentally informed decisions.

## 7. Future Coating Systems

### 7.1. Lamellar Fillers in Primers

A Zn primer layer provides galvanic corrosion protection to steel only through the Zn particles. The Zn needs to be electrically connected to the steel to give galvanic cathodic protection; however, Zn that is not connected can still help by consuming oxygen that is trying to diffuse through the coating before it gets to the metal surface to cause corrosion. Adding 1% mass conductive polymers (polyaniline) can act as Zn activators to allow more Zn to participate in galvanic corrosion protection. The activators can also "self-heal", by which the polymer can rebuild itself because of its less-rigid structure [30–32]. One material that has shown effectiveness in performing the described function is stainless steel. Stainless-steel flakes (SSF) with diameters of 26 μm have long, flat, lamellar structures and can act as a barrier to corrosive species as well as conductors for electrically connecting more zinc to the steel to provide cathodic protection. The addition of 2.5% weight SSF in a zinc primer layer maintained cathodic protection longer than did pure zinc coatings and enhanced barrier properties once all the electrically connected zinc was preferentially corroded [33].

Another conductive filling that has received a great deal of attention is fullerene carbon nanotubes (CNTs). Tesla Nanocoatings has led the innovations in this field, where joint research with the DoD in the USA has produced a two-layer coating system incorporating CNTs that has outperformed all other DoD-considered coating systems. The main improvement to the primer was that the CNTs made the layer electrically conductive, allowing the primer to contain a lower weight% of zinc but with the same amount of sacrificial zinc electrically connected to the corroding surface as would be found in a higher weight% zinc primer. The lower zinc content and increased CNTs created a flexible, abrasion-resistant layer. Tesla Nanocoatings finished their primer coating system with a single epoxy topcoat to create a two-coat system that outperforms other three-coat systems.

To utilize a new coating system in a structural bridge application, most coating approval procedures such as NEPCOAT's require a proven track record of coating system performance before product approval. However, the FHWA has an Accelerating Innovation

Deployment (AID) initiative for developing innovations in highway transportation. To date, this initiative has given over USD 80 million for new technologies. Tesla Nanocoating's two-coating system was selected by Missouri's DOT to be used in the field on a bridge in 2016. Even though the paint was more expensive, the labor costs were predicted to be reduced by 25%–50%, which also corresponds to less impact on quality of service for the bridge while it is partially closed for maintenance.

### 7.2. Self-Stratifying Coatings

Self-stratifying coatings are single-spray coatings that will separate into two layers because they are unstable emulsions of incompatible polymer blends, as demonstrated in Figure 5 [34].

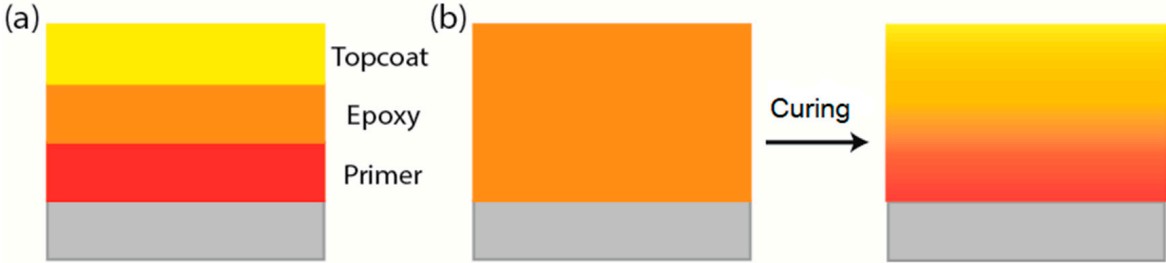

**Figure 5.** (**a**) A normal three-coat system; (**b**) a self-stratifying coating system as sprayed (**left**) and drying over time (**right**).

Both water-borne and solvent-borne coating systems can be stratified; pigments such as titanium dioxide and lamellar iron oxide can be included as well to achieve coating thicknesses of up to 500 μm. This technology has the potential to significantly reduce the solvent and volatile organic compounds (VOCs) used in the deposition, the labor costs used, and the risks of deadhesion between coatings by reducing internal stresses.

Research into self-stratifying coatings is intensifying; researchers at the Hempel Foundation Coatings Science and Technology Center (CoaST) have developed predictive tools to screen potential self-stratifying coating compositions [35]. For this model, they input the Hansen solubility parameters of polymers to find two appropriate polymers, model rate-based kinetics to find appropriate hardeners for the system, and individually select binary solvent mixtures to develop a potential library of self-stratifying coatings, which includes mainly epoxy/topcoat coating systems.

Despite these technological advances, and to the best of our knowledge, these products have not been used on bridges. Self-stratifying coating systems, especially thicker coating systems, are at risk of retaining solvent, which would drastically reduce the lifetime and durability of the coating system. The curing procedures are more sensitive to surface preparation and temperature than the other coating systems discussed before this section, and highly trained coating system experts may be required for coating bridges with self-stratifying coating systems.

### 7.3. Hydrophobic and Super-Hydrophobic Coatings

Some new coating technologies focus on improving the barrier properties of the top layer of a coating system by improving the hydrophobicity of the coating. A hydrophobic coating is roughly defined as having a surface contact angle of greater than 90 degrees between a water droplet and the coating. Increasing hydrophobicity is desirable because it greatly retards the transport of water (which is necessary for corrosion to occur) to the steel structures by isolating water droplets on a surface, quickly removing droplets that fall on the surface, or preventing contact between solutions and the coating by trapping air between unique coating surface geometries and the solution [36–39]. Higher hydrophobicity coatings are currently in use, such as polysiloxane and polyurethane topcoats, that each have approximately 90-degree contact angles, but further advances are possible.

One way to achieve higher hydrophobicity is through the incorporation of <20 nm sized $SiO_2$ nanoparticles modified with polydimethylsiloxane (PDMS) [40]. Roshan et al. [40] found that incorporating these nanoparticles in 4 wt% concentrations into polyurethane films enhanced the surface contact angle from 92 degrees (without nanoparticles) to >160 degrees. This superhydrophobic coating increased internal film resistance over 8 weeks of submerged testing, demonstrating that water was very slowly diffusing into the film. However, for commercial topcoat applications, a fine balance between UV resistance, friction co-efficient, and adhesion strength must be found.

### 7.4. Smart Coating Systems with Nanoparticles

Some new coating systems are being developed with micro- or nanoparticles that encapsulate chemical inhibitors to increase the coating system's corrosion-inhibiting properties. Some coating systems also contain nanoparticles that encapsulate visual indicators of degradation that can aid in the maintenance scheduling of bridges. Upon the action of external factors, such as the presence of moisture or corrosion products in the layer or the accumulation of abrasions, on the coating system, these nanoparticles break apart to release the desired inhibitor/indicator. These nanoparticles are produced by interfacial polymerization or emulsion polymerization, which generally involves mixing two immiscible liquids containing polymerizing agents and the inhibitors or indicators to be encapsulated. These particles are then cured and added to coating systems in concentrations as low as 3 weight% or as high as 50 weight%.

In fact, several patents for pH-sensitive microencapsulation of pH indicators were filed as part of the corrosion-mitigation initiative at the National Aeronautics and Space Administration (NASA) [41–43]. NASA desired a quick way to check the integrity of bolts and faying surfaces without performing advanced analysis techniques or deconstructing the asset. The release of pH indicators in response to local changes in pH that are caused by corrosion processes will identify when corrosion has started and would be visible with the naked eye, allowing advanced planning of maintenance and rehabilitation of assets. However, there are some obstacles for nanoparticles to overcome, such as the price of synthesis of nanoparticles, the release of nanoparticles into the environment, and the formation of inclusions of nanoparticles, which can lead to premature coating failure.

### 7.5. Layered-Double Hydroxides

Layered-double hydroxides (LDH) are 2D materials, with the most common LDH being made of magnesium, aluminum, and hydroxides [44]. The tunability of the chemical composition and shape of LDH allows applications in water purification, electrochemical energy storage through conductive 2D networks, superhydrophobic films, pharmaceuticals in drug delivery, specialty catalysts, and in corrosion protection, where corrosion inhibitors can be stored inside the 2D structure. LDH technology for coating systems can be used similarly to nanoparticles to deliver corrosion inhibitors and corrosion indicators but also as protective films, where they primarily act as a barrier to prevent chlorides, water, and oxygen from reaching the metal's surface.

There is a nearly endless list of LDH examples and desirable properties to discuss for steel corrosion, where research on LDH technology is thought to have begun in 1834. One study incorporated two types of LDH ($Zn$-$Al$-$PO_4{}^{3-}$ and $Zn$-$Al$-$NO_3{}^-$) into a silane primer at about 12 weight% and added an epoxy/polyamide topcoat [45]. Alibashki et al. [45] compared the LDH-containing primers to primers without LDH and, through salt-spray and adhesion testing, found that the silane primer with $Zn$-$Al$-$PO_4{}^{3-}$ performed much better than primers without them. When the coating failed, zinc phosphate ($Zn_3(PO_4)_2$) and iron phosphate ($Fe(PO_4)$) were observed to form a film on the surface of the exposed steel; both chemicals are known to inhibit corrosion through the formation of a strong film [46].

Similar to nanoparticles, the main negative aspects of LDH are the impact to life when LDH are released into the water and inclusions that may result from LDH accumulating

due to unforeseen application environments. Inclusions could increase the failure rate by preferentially causing cracking due to large internal stresses placed on the films.

## 8. Concluding Remarks

1. ISO atmospheric corrosivity of regions in the United States and Canada was investigated for the coating selection; most locations near oceans show severe C4–C5 corrosivity, while most inland locations have lighter C1–C2 levels. Additionally, micro-corrosivity on bridge elements was discussed in locations with high relative humidity and use of de-icing road salts;

2. Road salts, pollution, and humidity are known to have adverse effects on steel corrosion, coating performance, and coating application procedures. Some areas of a bridge are exposed to more of these conditions than others. The most robust coating systems must be applied here, and maintenance in these areas will be more frequent. The optimal maintenance scheduling for a bridge likely consists of not performing maintenance on the entire bridge at once;

3. Zinc-rich primer coatings are the most commonly used coating systems for structural steel bridge protection; however, detailed life-cycle cost analysis and maintenance scheduling should be considered for various bridge locations to select the best-performing system for their specific conditions. Metallizing and hot-dipped galvanizing are high-performing options with a significantly higher upfront cost and are best used in corrosive environments that are hard to access and maintain. Non-zinc coating systems have not shown performance levels as high as zinc-based systems and are only used in less-corrosive environments;

4. The environmental impacts of available coating systems need to be quantified for decision making and should be investigated in the future. Ideally, life-cycle cost assessments and LCA should be conducted in tandem for bridge designs so that bridge engineers are aware of the GWP of their decisions;

5. The coatings industry is evolving, and new coating systems that can influence the corrosion protection of steel bridges are being introduced. For example, smart coatings can influence maintenance scheduling by providing visible information that marks corrosion initiation. Robust systems that are applicable in most environments would reduce the worker time required to apply multiple coatings, thereby impacting labor costs or maintenance scheduling and corrosion detection.

**Author Contributions:** Data curation, J.R.A. and F.J.; formal analysis, N.E., J.R.A. and F.J.; investigation, N.E. and J.R.A.; methodology, N.E., J.R.A. and J.J.N.; software, F.J.; supervision, N.E. and J.Z.; validation, J.Z.; writing—original draft, N.E. and J.R.A.; writing—review and editing, J.Z., F.J. and J.J.N. All authors have read and agreed to the published version of the manuscript.

**Funding:** The authors acknowledge the financial support of Infrastructure Canada to the NRC's Climate Resilient Built Environ-ment (CRBE) Initiative. The study presented herein is executed under the CRBE Initiative.

**Institutional Review Board Statement:** Not applicable.

**Informed Consent Statement:** Not applicable.

**Data Availability Statement:** Some or all data, models, or code that support the findings of this study are available from the corresponding author upon reasonable request.

**Conflicts of Interest:** The authors declare no conflict of interest.

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
