# Peer review of "Steel Bridge-Coating Systems and Their Environmental Impacts: Current Practices and Future Trends"

_coatings, doi:10.3390/coatings13050850_

Round 1
Reviewer 1 Report
Sentence 25: “This physical barrier generally stops chlorides from ….”. Passive layers as mentioned in the manuscript as “an invisible oxide consisting of Cr(III) and Cr(VI) on 24 stainless steel” do not stop reaching corrosive species to the substrate surface, rather they reduce the corrosion rate by controlling the diffusion of anions and/or metal cations.
Sentence 155: There is no need to have a numbering “v”.
Some advanced coating technology for bridge application such as Hydrophobic and super-hydrophobic coatings are missing.
Some corrections have to be done in References:
Ref. 1. Standard number should be given as “ISO 12944”.
Some addresses such as Ref 28 and 29 are not complete.
Ref. 10 and Ref. 26 are the same.
Ref. 22 and Ref. 23 are the same.
Author Response
We would like to express our gratitude to the reviewers for their time and insightful comments on our paper. Your valuable feedback and suggestions have tremendously improved the quality of our work and helped us to identify and address the areas that needed improvement. Below is the detailed answer to the comments:
Reviewer 1:
Sentence 25: “This physical barrier generally stops chlorides from ….”. Passive layers as mentioned in the manuscript as “an invisible oxide consisting of Cr(III) and Cr(VI) on 24 stainless steel” do not stop reaching corrosive species to the substrate surface, rather they reduce the corrosion rate by controlling the diffusion of anions and/or metal cations.
- The suggestion has been addressed in line 28
Sentence 155: There is no need to have a numbering “v”.
- Numbering V is removed
Some advanced coating technology for bridge application such as Hydrophobic and super-hydrophobic coatings are missing.
- A new section titled “Hydrophobic and super-hydrophobic coatings” is added to the manuscript
Some corrections have to be done in References:
Ref. 1. Standard number should be given as “ISO 12944”.
Some addresses such as Ref 28 and 29 are not complete.
Ref. 10 and Ref. 26 are the same.
Ref. 22 and Ref. 23 are the same.
- References were checked and corrected.
Reviewer 2 Report
The Manuscript entitled "Steel Bridge Coating Systems and Their Environmental Impacts: Current Practices and Future Trends" covers very interesting topic.
The Manuscripts is within the scope of the journal and is valuable scientific work. After revising some minor issues it can be published:
- Authors should present in comparative manner with more detailed description about these categories.
Maybe table with compounds composition and concentrations. for Categories.
- Implementing some data into this figure would be beneficial. (Figure 2).
- There is need to explain how these chemical acid environment affects the durability of epoxy resin. what are the destruction processes. It will let to understand what further steps should have been done.
Maybe some potential reactions that occurs.
- quality of figure 3 can be improved
Author Response
We would like to express our gratitude to the reviewers for their time and insightful comments on our paper. Your valuable feedback and suggestions have tremendously improved the quality of our work and helped us to identify and address the areas that needed improvement. Below is the detailed answer to the comments:
Authors should present in comparative manner with more detailed description about these categories.
Maybe table with compounds composition and concentrations. for Categories.
- More detail is added to the manuscript (Line 110 to 130)
- Implementing some data into this figure would be beneficial. (Figure 2).
- Figure 2 is updated
- There is need to explain how these chemical acid environment affects the durability of epoxy resin. what are the destruction processes. It will let to understand what further steps should have been done.
Maybe some potential reactions that occurs.
- More detail is added to the manuscript including a new figure (Line 217 to 246)
- quality of figure 3 can be improved
- Figure 3 (now figure 4 is updated).
Reviewer 3 Report
The article regarding Steel Bridge Coating System is interesting.
I have some suggestions:
In the introduction, it could be mentioned that there are two categories of non-invasive corrosion prevention techniques are applied for steel bridge corrosion protection: (1) surface modification or surface treatment, (such as Paint Coatings, Galvanized Coatings; Thermal Spray Metallic Coatings); and (2) alternative steel modifications, like weathering and stainless steels. There are explanations at "Coating Systems Selection" part, but maybe there are useful to be mentioned also in Introduction.
Life cycle cost analysis part with Table 2 showing "Performance matrix for selecting bridge coating systems" maybe could be more clear presented. The table present more more feasible solutions for each case? Because it is only one cost for each type of structure.
Author Response
We would like to express our gratitude to the reviewers for their time and insightful comments on our paper. Your valuable feedback and suggestions have tremendously improved the quality of our work and helped us to identify and address the areas that needed improvement. Below is the detailed answer to the comments:
Reviewer 3:
In the introduction, it could be mentioned that there are two categories of non-invasive corrosion prevention techniques are applied for steel bridge corrosion protection: (1) surface modification or surface treatment, (such as Paint Coatings, Galvanized Coatings; Thermal Spray Metallic Coatings); and (2) alternative steel modifications, like weathering and stainless steels. There are explanations at "Coating Systems Selection" part, but maybe there are useful to be mentioned also in Introduction.
- Thank you, additional text was added to introduction (Line 32 to 34).
Life cycle cost analysis part with Table 2 showing "Performance matrix for selecting bridge coating systems" maybe could be more clear presented. The table present more more feasible solutions for each case? Because it is only one cost for each type of structure.
- Thank you for your comments, the reference 24 covers this subject in more detail. The focus of this paper is more the environmental life cycle analysis and further details on the various life cycle cost options is out of the scope of this study.